# Bariatric Surgery and Hypertension

**DOI:** 10.3390/jcm10184049

**Published:** 2021-09-07

**Authors:** Elisenda Climent, Anna Oliveras, Juan Pedro-Botet, Albert Goday, David Benaiges

**Affiliations:** 1Endocrinology Department, Hospital Universitari del Mar, 08003 Barcelona, Spain; ecliment@parcdesalutmar.cat (E.C.); 86620@parcdesalutmar.cat (J.P.-B.); agoday@parcdesalutmar.cat (A.G.); 2IMIM (Hospital del Mar Medical Research Institute), 08003 Barcelona, Spain; 87052@parcdesalutmar.cat; 3Department of Medicine, Universitat Autònoma de Barcelona, 08139 Barcelona, Spain; 4Nephrology Department, Hospital Universitari del Mar, 08003 Barcelona, Spain; 5Department of Experimental and Health Sciences, Area of Medicine, Universitat Pompeu Fabra, 08002 Barcelona, Spain; 6Red de Investigación Renal (REDINREN), Instituto Carlos III-FEDER, 28029 Madrid, Spain; 7Centro de Investigaciones Biomédicas en Red de Obesidad y Nutrición, CIBERobn, 28029 Madrid, Spain; 8Consorci Sanitari de l’Alt Penedès i Garraf, 08720 Vilafranca del Penedès, Spain

**Keywords:** bariatric surgery, obesity, severe obesity, hypertension, blood pressure, modifications of structural changes

## Abstract

A clear pathogenetic association exists between obesity and arterial hypertension, becoming even more evident in subjects with severe obesity. Bariatric surgery has proved to be the most effective treatment for severe obesity, with its benefits going beyond weight loss. The present review aimed to determine the effects of bariatric surgery on arterial hypertension evident in short- and long-term follow-ups. Moreover, the differences between surgical techniques regarding hypertension remission are described as well as the possible pathophysiologic mechanisms involved. In addition, the effects of bariatric surgery beyond blood pressure normalization are also analyzed, including those on target organs and cardiovascular morbidity and mortality.

## 1. Introduction

Hypertension (HTN) is one of the best known and most widely studied cardiovascular risk factors, and a close correlation between obesity and HTN has been extensively demonstrated [1]. Thus, HTN prevalence in subjects with obesity varies between 60 and 77%, and it is clearly higher than the 34% observed in subjects with normal weight [2]. The mechanisms by which obesity raises the risk of developing HTN are multifactorial, involving structural, functional, and hemodynamic changes in the cardiovascular system [3].

Conventional medical treatment for morbid obesity has previously achieved mild outcomes, which are probably related to limited long-term adherence to lifestyle modifications in some patients [1]. By contrast, bariatric surgery (BS) has proved to be the most effective therapy for these patients when both weight loss and comorbidity remission after surgery, including HTN, were evaluated [2,3].

In this respect, owing to the widely known systematic review published by Buchwald et al. [3] in 2004, which included a total of 22,094 patients, it has been accepted that approximately three of every five subjects undergoing BS achieve HTN remission. However, it must be considered that this meta-analysis mainly included studies with a short-term follow-up, with the surgical procedures performed at that time (gastric bypass (GB), gastric band, and biliopancreatic diversion), and most studies were retrospective and with great heterogeneity regarding HTN remission definition. In recent years, several prospective studies have reported mid- and long-term results after surgery, with laparoscopic sleeve gastrectomy (LSG) emerging as the most used BS technique worldwide [4]. Moreover, the possible underlying mechanisms responsible for HTN improvement after BS have been further evaluated, together with the possible benefits beyond weight loss. The present narrative review aimed to delve into these newly acquired data.

## 2. Bariatric Surgery Effects on Blood Pressure

### 2.1. Short-Term Effects on Blood Pressure

HTN remission in the short term (<3 years) after BS has been widely analyzed in observational studies, some meta-analyses, and a few randomized controlled trials (RCT) [5,6,7,8,9]. Schiavon et al. [10] in 2018 published the first RCT specifically aimed at evaluating the effect of BS on HTN remission. The GATEWAY (Gastric Bypass to Treat Obese Patients with Steady Hypertension) trial [10] included patients with HTN (using ≥2 medications at maximum doses or >2 at moderate doses) and a body mass index between 30.0 and 39.9 kg/m^2^. Subjects were randomized to GB plus medical therapy or medical therapy alone. The primary endpoint (≥30% reduction in the total number of antihypertensive medications while maintaining systolic and diastolic blood pressure < 140 and 90 mmHg, respectively, at 12 months) occurred more frequently in the GB group (83.7%) compared to the control group (12.8%). Moreover, HTN remission 1 year after surgery, defined as systolic and diastolic blood pressure < 140 and 90 mmHg, respectively, with previous withdrawal of all medication, occurred in approximately one-half of the patients in the GB group and none in the conventional treatment group. It is noteworthy that the HTN remission rate after BS obtained in the GATEWAY trial [10] was lower than those described in other previous reports [5,6,9], including the Buchwald et al. meta-analysis [3]. This was probably due to the first including patients who required an “aggressive” antihypertensive treatment, in comparison to the other studies where the included patients needed one or no antihypertensive medication. Hence, taking these results into account, if BS were primarily indicated to control refractory HTN, the chance of achieving remission would probably be close to 50% in the short term. In accordance with these data, it has been reported that the number of antihypertensive drugs prior to surgery was associated with a lower remission rate during the first year [9]. Another relevant result obtained from the GATEWAY study was that no differences in systolic and diastolic blood pressure levels were observed between groups during follow-up. This seems to indicate that if good titration of the medication is made during follow-up considering blood pressure levels, the effects of BS on HTN are mainly reflected in the reduction in the number of antihypertensive medications.

### 2.2. Mid- and Long-Term Effects

Less evidence exists on the mid- (3–5 years) and long-term (>5 years) effects of BS on HTN remission compared to other obesity comorbidities such as type 2 diabetes, and this evidence is mainly available from observational studies [2,11,12].

The results obtained in the mid- and long-term after BS were more modest compared to those achieved with a shorter-term follow-up. Regarding this, our group had previously evaluated HTN remission after BS with a 36-month follow-up, observing that 68.1% of hypertensive patients showed HTN remission 1 year after the surgical procedure, 21.9% of whom had relapsed at 3 years [9]. A possible justification for these less favorable results seems to be explained, at least in part, by weight regain after surgery. It must be taken into account that maximum weight loss is achieved during the first 12 months post-surgery, and from this point onwards, weight regain and worsening of certain metabolic parameters usually emerge. This coincides with the results obtained in our cohort, where milder weight loss during the first year was also associated with increased HTN recurrence at 3 years [9].

However, BS still presents more beneficial outcomes in the mid- and long-term follow-up compared to conventional treatment. In this respect, various RCT [13,14,15] compared BS to conventional treatment with a 5-year follow-up. Mingrone et al. [13] found that the BS group and conventional treatment maintained similar blood pressure levels 60 months after surgery. Nevertheless, more subjects in the latter group required antihypertensive medication (73% with conventional treatment versus 58% after GB and 32% after biliopancreatic diversion). Similarly, Ikramuddin et al. [15] also found a favorable trend toward BS. In that study, primary systolic blood pressure < 130 mm Hg at 5 years was obtained in 73% in the GB group versus 49% in the lifestyle and intensive medical management group (odds ratio (OR), 2.71; 95% CI, 0.95–7.78; *p* = 0.06).

The superior results obtained with a surgical approach compared to lifestyle modifications have also been further confirmed with a longer-term follow-up. In this respect, the Swedish Obese Subjects cohort [2] observed a greater reduction in blood pressure levels after GB compared to a non-surgical approach, with a mean follow-up of 10 years. Moreover, the percentage of patients requiring antihypertensive treatment was also lower after BS compared to the control group (35% vs. 53%; *p* < 0.001), with these results being in line with other previous studies [11,16].

Systematic reviews and meta-analyses also confirmed the superiority of BS, which was previously observed with a short-term follow-up. In this respect, Vest et al. [17] in 2012 (including 70 observational studies and three RCT) reported a 63% resolution or improvement in HTN with a mean follow-up of approximately 5 years. Similarly, Wilhelm et al. [8] in 2014 (including 31 prospective and 26 retrospective studies) observed 50% and 63.7% HTN resolution or improvement, respectively, with a mean follow-up varying from 1 week to 7 years post-surgery. Of the 57 studies included, 32 reported HTN improvement (OR, 13.24; 95% CI, 7.73–22.68; *p* < 0.00001) and 46 reported HTN resolution (OR, 1.70; 95% CI, 1.13–2.58; *p* = 0.01).

However, although studies with a longer follow-up confirmed the beneficial outcomes after BS in comparison to conventional treatment regarding HTN evolution, an RCT specifically focused on evaluating HTN remission at mid- and long-term after BS is lacking. Moreover, the possible differences among the most used surgical procedures (including malabsortive, restrictive, or both surgical approaches) must not be ignored, as detailed below.

### 2.3. Differences among Surgical Procedures

Considering the different BS procedures, GB has been considered, until recently, the gold standard technique owing to its favorable results in both weight loss and comorbidity remission [18]. However, in recent years, LSG also proved to achieve comparable promising results to GB, hence becoming the most used BS procedure in 2014 [4]. Moreover, LSG is a technically easier procedure compared to GB [19,20], with a presumably lower risk of perioperative complications [18].

In order to shed light on the effects of both BS techniques, our group carried out a meta-analysis to evaluate 1 and 5-year HTN remission after both procedures [21]. Thirty-two articles were involved, with a higher HTN remission rate being observed with GB compared to LSG both at 1 year (RR, 1.14, 95% CI, 1.06–1.21) and at 5 years (RR, 1.26, 95% CI, 1.07–1.48) after surgery. Blood pressure improvement after surgery was also evaluated. No differences were found between GB and LSG in terms of systolic or diastolic blood pressure changes at both 1 and 5 years. Thus, we could speculate that although patients in the LSG group were less likely to present HTN remission after BS, and hence not all the antihypertensive medication could be withdrawn, overall blood pressure levels in both groups were equivalent after surgery. It is also important to highlight the fact that the superiority of GB over LSG was observed when all studies were included, as well as when only the highest evidence studies (RCT) were evaluated.

Thus, although some studies obtained more promising results regarding HTN remission after GB compared to LSG, the superiority of GB must be further confirmed with longer-term follow-up (>5 years).

### 2.4. Metabolic Surgery and HTN

Owing to the favorable results (which go beyond weight loss) of BS in obese subjects, the concept of metabolic surgery has gained importance in recent years [22], with the focus on the physiologic modifications that occur after surgery, which lead to comorbidity improvement [23]. Moreover, the metabolic effects of the surgical procedure become more evident when obesity comorbidities improve within days after BS and when significant weight loss has not yet been achieved [8].

This fact has opened debate on whether BS should be indicated in patients with body mass index < 35 kg/m^2^ for comorbidity improvement, which was addressed in previous observational publications mainly aimed at glycemic improvement after surgery but also at achieving hopeful results regarding HTN remission [24,25].

Five RCT [5,26,27,28,29] also assessed the effects of BS in subjects with class I obesity, observing positive results in blood pressure evolution, nearly equivalent to those obtained in patients with body mass index > 35 kg/m^2^ (Table 1). However, the main limitation when evaluating these data was the heterogeneity of the definitions used for remission or improvement in the different studies, as some considered total withdrawal of antihypertensive medication and others only blood pressure normalization. In order to standardize all studies evaluating comorbidity remission with grade I obesity, the International Federation for the Surgery of Obesity and Metabolic Disorders (IFSO) realized a position statement in 2014 [30] summarizing the scientific background concerning BS in class I obesity. They concluded that a clinical decision of whether to deny BS to these patients should be based on a more comprehensive evaluation of the patient’s current global health and on a more reliable prediction of future morbidity and mortality. Hence, future observational studies and RCT with a longer-term follow-up are necessary.

### 2.5. Possible Mechanisms Related to HTN Improvement

Although weight loss has proved to be a key factor in comorbidity improvement after BS, other underlying factors may also play an important role. With regard to blood pressure improvement after BS, the reasons are probably multifactorial and remain under debate (Figure 1) [31,32].

It has been speculated that a decreased inflammatory response together with an improvement in insulin resistance could reduce arterial stiffness and sodium reabsorption and hence lead to normalization of blood pressure levels [33]. Patients with central obesity are known to have increased activation of the renin–angiotensin–aldosterone system, which may also normalize after surgery [34].

In addition, an increase in gastrointestinal gut hormones such as peptide YY (PYY) and glucagon-like peptide-1 (GLP-1) could also play an important part due to their effects on the gastrointestinal system together with a diuretic and natriuretic effect on the kidney [35]. Furthermore, a possible effect of GLP-1 on the sympathetic nervous system, which may play a part in the blood pressure-lowering effect after BS, has also been described [36]. Ghrelin may also aid in normalizing blood pressure levels, although its levels may raise, fall, or remain unchanged after BS, depending on the surgical procedure [37].

Furthermore, adipokines and other inflammatory cytokines also appear to be related to HTN recovery. In this respect, previous studies observed a decline in leptin levels from 1 week up to 1 year after BS together with increasing adiponectin concentrations [38]. Moreover, as insulin sensitivity increases, C-reactive protein and interleukin-6 levels decrease, thus ameliorating adipocyte inflammation and in turn preventing vascular constriction [39].

Finally, the resolution of other obesity comorbidities (which share pathophysiologic mechanisms with HTN) such as obstructive sleep apnea could also play a part in blood pressure improvement [40,41].

The underlying mechanisms related to the possible superiority of GB over LSG are also worth mentioning. The main accepted hypothesis is that these differences could be explained by the superior weight loss after GB in the mid- and long-term follow-up. As mentioned previously, the possible role of gastrointestinal hormones in HTN improvement after surgery gains value, as some studies observed a decrease in blood pressure levels within the first week post-BS and when weight loss was minimal [8]. In this respect, a previous study found significant reductions in both systolic (9 mm Hg) and diastolic (7 mm Hg) blood pressure 1 week after GB, and these were maintained 1 year after surgery [42]. Considering the different surgical procedures, GLP-1 and PYY are known to increase after both, but they increase more intensely after GB [42,43], which may account for the more favorable results after this procedure.

## 3. Bariatric Surgery Benefits beyond Blood Pressure Improvement

### 3.1. Organ Damage Changes

Patients with morbid obesity have a higher prevalence of target organ damage than patients of normal weight, and HTN is clearly related to its development. These target organ alterations mostly refer to changes in heart, vessels, and kidney structure and function [44].

#### 3.1.1. Cardiac Changes

Regarding cardiac changes, several works reported echocardiographic alterations, both morphologic and functional, in obese patients [45,46]. The main alterations consisted of left ventricular (LV) hypertrophy and impaired LV diastolic function, while LV systolic dysfunction was less common and, on these lines, reports concerning the ejection fraction in obese patients were contradictory [47]. Morphologic LV alterations have been described in patients with morbid obesity, with 56% of LV hypertrophy being reported from a meta-analysis of 22 studies including 5486 obese subjects [45]. Many of these changes are precursors of more overt forms of cardiac dysfunction and heart failure [48]. Indeed, obesity clearly increases the risk of atrial fibrillation, myocardial infarction, heart failure, and sudden death [49]. Beyond findings from observational epidemiology, Larsson et al. [50] recently found evidence that a genetically instrumented 1 kg/m^2^ higher body mass index is associated with an increased risk of aortic stenosis, heart failure, deep venous thrombosis, HTN, peripheral artery disease, coronary artery disease, atrial fibrillation, and pulmonary embolism (estimates in the range of 6–13% higher risk). The findings for fat mass were broadly consistent. Specifically, the link between obesity and heart failure is known to be stronger than those for other cardiovascular disease subtypes and is uniquely unexplained by traditional risk factors [51]. However, the findings apparently diverged from observational studies for ischemic stroke, and this field merits further investigation [50].

In relation to the mechanisms responsible for cardiac improvement after BS, several authors concur in that the effects of weight-loss surgery on cardiac function and morphology are either hormonally or centrally regulated, probably with an important role for leptin and other adipokines [52], as well as for the renin–angiotensin–aldosterone axis [53]; however, further insight needs to be gained into the mechanisms underlying changes in cardiovascular function after weight loss.

Importantly, these cardiovascular structure and function alterations have also proved to be reversible with weight loss strategies such as BS, resulting in lowered cardiovascular risk [54]. The effects of BS on cardiac structure and function were recorded in a systematic review of 23 studies and meta-analysis [55], showing that in obese patients with preserved LV systolic function, BS induced significant decrements of absolute LV mass and relative wall thickness (RWT), which are all reliable indexes of LV hypertrophy and LV geometry that have been shown to predict cardiovascular outcomes. Furthermore, that meta-analysis showed improvements in LV diastolic function, as reflected by a clear-cut increase in the mitral flow ratio of the early (E) to late (A) ventricular filling velocities (E/A ratio), as well as decreases in left atrium size, which is an indirect marker of chronically elevated LV filling pressure and diastolic dysfunction. As for LV hypertrophy and RWT, similar results were reported by Owan et al. [56] 2 years after BS. Those authors found that the decreases in LV mass index and RWT correlated with body mass index reduction but not with changes in blood pressure. Of note, one of the most salient observations of the BARIHTA study by our group was that even severely-obese patients with strictly normal blood pressure experience an improvement in morphologic and functional LV parameters after BS [53].

#### 3.1.2. Vessel Changes

One of the main manifestations of vessel alteration is the development of arterial stiffness (AS). It is considered to be an independent cardiovascular risk factor [57] and is defined as the diminished ability of an artery to expand and contract in response to a given pressure change [58]. Pulse wave velocity (PWV) is the gold standard for AS measurement [59]. In the last two decades, excess body weight has been found to be associated with greater aortic stiffness in young and older adults [60]. Therefore, increased AS may be one of the mechanisms by which obesity raises cardiovascular risk independently of traditional risk factors. Indeed, high PWV predicts outcomes independent of the Framingham Risk Score, and it is associated with increased cardiovascular disease risk regardless of HTN status [61]. On the same lines, some authors suggested that AS may precede rises in systolic blood pressure and incident HTN in obese individuals [62].

Regarding the effect of BS on AS, several studies reported a significant decrease in both PWV and the augmentation index, another marker of AS, several months after BS [63,64,65]. The potential mechanisms responsible for the reduction in AS after weight loss are not clear. Some authors [66] found a correlation between weight loss and reduction in PWV independently of changes in established hemodynamic and cardiometabolic risk factors, and other groups [64], but not all [60], suggested that this correlation is mediated by the drop in blood pressure. On the other hand, elevated cardiac volume and output in obese individuals were also noted as possible mediators of AS, more importantly than elevated BP [67].

#### 3.1.3. Renal Changes

Obesity is an independent risk factor for kidney disease, regardless of diabetes and HTN, both of which are driven largely by obesity [68]. Hyperfiltration is the hallmark of obesity-associated kidney dysfunction, and the main proposed mechanisms for this association are hemodynamic factors, inflammatory cytokines, and renal lipotoxicity [68,69]. As regards hemodynamic factors [70], excessive weight initially causes functional renal vasodilation and increases in renal blood flow and glomerular hyperfiltration prior to nephron injury. These changes are later followed by declines in renal blood flow and the glomerular filtration rate (GFR) as a result of kidney injury and gradual loss of nephrons. Increased extracellular fluid volume results from the obesity-associated increase in tubular sodium reabsorption. This may be related to the elevated levels of anti-natriuretic hormones such as angiotensin II and aldosterone, as a consequence of both kidney compression by visceral, perirenal, and renal sinus fat and of the increased renal sympathetic nerve activity. These and other contributors may be linked by the altered macula densa feedback (tubuloglomerular feedback) to the observed afferent arteriola vasodilation. Sodium balance may be re-established despite increased sodium chloride reabsorption in the loop of Henle through compensatory increases in the GFR and blood pressure elevation. Furthermore, mineralocorticoid receptor (MR) activation may also contribute to renal vasodilation. MR expressed on macula densa cells are activated by aldosterone, thereby increasing their production of nitric oxide and leading to renal vasodilation and glomerular hyperfiltration. Despite the adaptative value of glomerular hyperfiltration in offsetting renal sodium reabsorption, this increase in glomerular hydrostatic pressure probably contributes greatly to the renal injury observed in obesity.

Obesity also favors a deleterious adipocytokine pattern [68,69] characterized by the overproduction of angiotensinogen and angiotensin II as well as the upregulation of pro-inflammatory cytokines such as interleukin-6, C-reactive protein, and tumor necrosis factor-α. These factors induce renal fibrosis via the transforming growth factor-β (TGF-β) pathway and via oxidative stress, as shown by experimental models. Moreover, obese individuals are known to have high levels of serum leptin and high expression of leptin receptors in the kidney, which also stimulate cellular proliferation and expression of the prosclerotic TGF-β1 cytokine implicated in the early scarring formation of renal failure. Finally, reduced levels of another adipokine, adiponectin, have been implicated as a mechanism of obesity-related renal impairment through podocyte damage leading to albuminuria. Pathologic changes due to long-lasting hyperfiltration include the development of glomerulomegaly and renal lesions of focal segmental glomerulosclerosis, leading to obesity-related glomerulopathy [71]. Thus, hyperfiltration, i.e., GFR higher than 120 mL/min/1.73 m^2^, and albuminuria, biomarkers of kidney function and damage, respectively, characterize renal alterations in obese patients.

The gold-standard method to assess the GFR is measurement of the renal clearance of an exogenous filtration tracer (inulin, 51 Cr-EDTA, 125 I-iothalamat, iohexol); however, most studies use GFR (eGFR) estimations derived from prediction equations. These equations were obtained by regression analyses in various populations with body mass index < 30 kg/m^2^, where the GFR was measured by the gold standard method, but these are not accurate in obesity classes II and III [68]. Thus, it is unclear how reliably creatinine-based eGFR equations perform among those with obesity, especially when faced with results normalized to a body surface area of 1.73 m^2^ since, after BS, patients lose not only fat but also muscle mass, which generates creatinine [72]. Furthermore, although body surface area, which is considered in the eGFR equations, is vastly reduced after BS, it is not reflected in the eGFR results routinely available [73]. Cystatin C has been suggested as a potential alternative since, unlike creatinine, it does not come strictly from muscle. However, it has not been validated as a reliable biomarker of GFR in obese patients, nor has its laboratory assay been standardized as for creatinine. On the other hand, measurement of albumin excretion rates via albumin-to-creatinine ratios (ACR) in fresh spot urines or absolute excretion rates in timed urine collection has become a more reliable measurement of renal damage [74].

Overall, patients with complicated obesity will likely benefit from the weight loss after BS [75]. Li et al. [76] reported a systematic review and meta-analysis from 32 studies showing significant reductions in hyperfiltration (measured GFR, eGFR, and creatinine clearance with and without adjustment for body surface area), albuminuria (defined as an ACR of more than 30 mg/g of creatinine), and proteinuria after BS. They reported a reduction in hyperfiltration (RR: 0.46, 95% CI 0.26–0.82, *p* = 0.008) after surgery when analyzed as a dichotomous variable as well as statistically significant decreases. Moreover, drops were observed in the incidences of albuminuria and proteinuria after BS of 58% and 69%, respectively (*p* < 0.0001 for both). Data on the 4047 patients included in the Swedish Obese Subjects study [77], comparing patients undergoing BS and controls followed up for a median time of 18 years, showed a lower incidence of chronic kidney disease (CKD) stages 4 and 5 in patients in the surgery group (adjusted HR = 0.33; 95% CI 0.18–0.62; *p* < 0.001). Similarly, O’Brien et al. [78] in a retrospective analysis reported a 59% lower incidence of nephropathy at 5 years in a cohort of 4000 diabetic patients undergoing BS compared to 11,000 matched non-surgically treated patients. Friedman et al. [75] analyzed 2144 obese patients who underwent BS and found an improvement in CKD risk categories in a large proportion of patients over a 7-year follow-up period. They reported that the reduction in risk was most pronounced in persons with high baseline risk.

As regards renal protective factors, Favre et al. [68] reported that low C-reactive protein levels, high fat mass, lack of HTN, and young age predicted kidney protection in severely obese patients undergoing BS.

The mechanisms behind the improvement in risk factors following BS are not well understood. Glomerular function may be related to restoration in homeostasis of the renin–angiotensin system through better renal perfusion and to the restitution of normal insulin signaling in glomerular podocytes and attenuation of hyperfiltration. Additionally, this improvement may also be secondary to reductions in the pro-inflammatory state related to obesity [74] as measured by urinary monocyte-chemoattractant protein-1/creatinine ratios [73]. It has recently been shown that the glucagon-like peptide 1 (GLP-1), an incretin hormone released by intestinal endocrine L cells, exerts renoprotective effects by inhibiting tubular reabsorption of sodium. These effects increase after BS, suggesting a role in the improvement in glomerular function. As for albuminuria remission, at least in obese diabetics, the restitution of podocyte health may be a key cellular event contributing to the benefits of BS.

Obese patients who undergo BS may also experience some renal complications. Lieske et al. [72] reported that up to 50% of these patients might be hyperoxaluric one year after surgery, and the risk of new kidney stone events doubled compared with unoperated obese controls. Nevertheless, the net effect on long-term kidney health is potentially positive for most patients.

### 3.2. Implications in Cardiovascular Morbidity and Mortality

Moving a step forward, the next question to answer is: what is the real impact of HTN improvement after BS in terms of cardiovascular morbidity and mortality reduction? It has previously been reported that BS reduces the number of cardiovascular events and mortality rates in patients with morbid obesity. For instance, the Swedish Obese Subjects Study Group [79] observed a reduced number of cardiovascular deaths in the surgical group compared to the control group (28 events among 2010 patients vs. 49 events among 2037 patients; adjusted hazard ratio (HR), 0.47; 95% CI, 0.29–0.76; *p* = 0.002) during a median follow-up of 14.7 years. In that same cohort, the number of total fatal or non-fatal cardiovascular events (myocardial infarction or stroke) was also lower in patients undergoing BS. Other studies yielded similar results, thereby confirming the beneficial effects of BS on cardiovascular morbidity and mortality [80,81].

Although the observed reduction in cardiovascular disease prevalence after BS is probably multifactorial, it can be assumed that HTN improvement probably plays a key role, although this remains to be confirmed. In fact, the Swedish Cohort [79] failed to find an association between weight loss and cardiovascular event reduction, thus highlighting the possible role of other factors that could explain the improvement in cardiovascular outcomes. In this respect, the decline in cardiovascular risk following the improvement in blood pressure levels after BS could be related to a reduction in target organ damage (including cardiac, vessel, and renal changes), as described previously in the present review [44,82].

However, the possible “obesity paradox” must also be acknowledged. This refers to a more favorable evolution regarding cardiovascular or renal outcomes in patients with a higher body mass index. This possible paradoxical effect observed in some studies could be explained by increased tumor necrosis factor (TNF-α) receptors in adipose tissue or an earlier diagnosis of cardiovascular events in the obese population, among others. Despite this, the underlying mechanisms of this possible paradox in obese population are still being investigated in order to achieve more solid conclusions [83].

## 4. Conclusions

BS has proved to be a highly effective treatment for obesity-associated HTN, achieving HTN remission in more than half of patients. However, a greater need for antihypertensive medication prior to BS and less weight loss during follow-up are both factors that may hinder the achievement of complete HTN remission.

Moreover, a decline in cardiovascular morbidity and mortality has also been observed after BS in morbidly obese subjects. These favorable results regarding cardiovascular outcomes may be mediated by multiple mechanisms that go beyond weight loss, one of which may be improved blood pressure levels together with a decline in target organ damage. However, future studies are required in this field for more solid conclusions to be drawn.

## Figures and Tables

**Figure 1 jcm-10-04049-f001:**
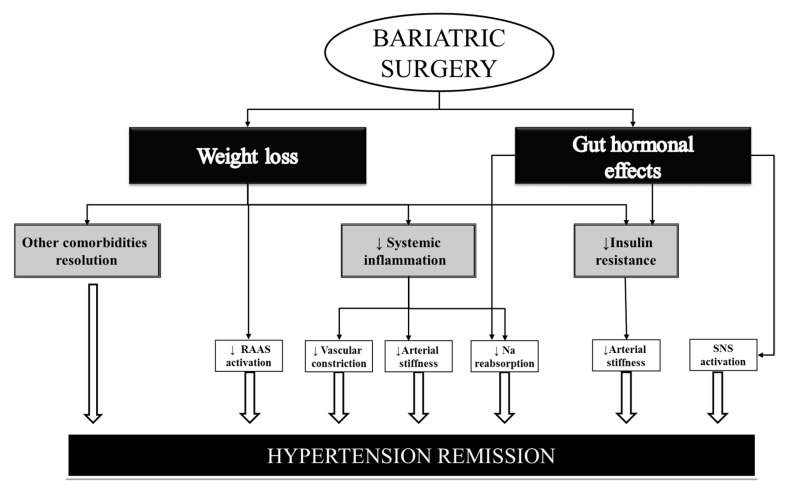
Mechanisms related to HTN remission. RAAS = Renin–angiotensin–aldosterone system; Na = Sodium; SNS ≠ sympathetic nervous system. ↑ ≠ increase; ↓ ≠ decrease.

**Table 1 jcm-10-04049-t001:** Randomized trials of bariatric surgery including patients with body mass index <35 kg/m^2^.

Study	*N*	BMI (kg/m^2^)	Follow-Up (Months)	Intervention Groups	Weight Loss	HTN-Related Outcomes
O’Brien et al.	80	30–35	24	LAGB Conventional therapy	87.2% EWL 21.8% EWL	−10.8% decrease in SBP/−10.9% decrease in DBP −7.2% decrease in SBP/−1.58% decrease in DBP
Dixon et al.	60	30–40 (21.7% BMI < 35)	24	LAGB Conventional therapy	20.7 TWL 1.7 TWL	−6.0 mmHg decrease in SBP/−0.7 mmHg decrease in DBP −1.7 mmHg decrease in SBP/−0.9 mmHg decrease in DBP
Lee et al.	60	25–35	12	Minigastric bypass LSG	94% EWL 76% EWL	12 months: SBP 119.6 mmHg/DBP 74.2 mmHg 12 months: SBP 123.5 mmHg/DBP 75.4 mmHg
Schauer et al.	150	27–43 (34% BMI < 35)	12	LRYGB LSG Intensive medical therapy	88% EWL 81% EWL 13% EWL	78% subjects antiHTN medication baseline/33% at 12 months 67% subjects antiHTN medication baseline/27% at 12 months 76% subjects antiHTN medication baseline/77% at 12 months
Ikramunddin et al.	120	30–40 (59.2% BMI < 35)	12	LRYGB Intensive medical therapy	26.1 TWL 7.9 TWL	Remission: 84% subjects SBP < 130 mmHg at 12 months 12 months: SBP 115 mmHg/DBP 68 mmHg Remission: 79% subjects SBP < 130 mmHg at 12 months 12 months: SBP 124 mmHg/DBP 74 mmHg

BMI: body mass index; DBP: diastolic blood pressure; EWL: excess weight loss; HTN: hypertension; LABG: laparoscopic adjustable gastric banding; LRYGB: laparoscopic Roux-en Y gastric bypass; LSG: laparoscopic sleeve gastrectomy; SBP: systolic blood pressure; TWL: total weight loss.

## Data Availability

Not applicable.

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
