# Peer review of "Bariatric Surgery and Hypertension"

_jcm, 2021, doi:10.3390/jcm10184049_

Round 1

Reviewer 1 Report

The present review aimed to determine the effects of  bariatric surgery on arterial hypertension in the short- and long-term follow-up. The literature review is carried out correctly.
No comments.

Author Response

We are grateful for the Reviewer's positive comments. Thank you. 

Reviewer 2 Report

The paper is clear and well written

Few minor remarks to be amended throughout the text:

  1. Lines 143, 168 and 245: "hybrid surgical procedure" this is unclear. Does it refer to absorption and restriction (improper) or to a robotic assisted laparoscopic approach?
  2. The meaning of some abbreviations is missing: e.g. line 289 ... mitral flow E/A (?) and line 397 ...CKD... Please double check text e fill the missing quotes
  3. Lines 190-196: this refers to a position statement as worldwide grade I obesity is not and indication for Bariatric Surgery and this should be amended
  4. Lines 428 and following: authors should add and discuss a short paragraph on the so called "obesity paradox", that reports quite the opposite.  

Author Response

Below are the responses to the Reviewer's comments:

  1. Lines 143, 168 and 245: "hybrid surgical procedure" this is unclear. Does it refer to absorption and restriction (improper) or to a robotic assisted laparoscopic approach? The term hybrid surgical procedure refers to a surgical procedure with a malabsortive and restrictive effect (for example, LRYGB) not a robotic assisted approach. Following the Reviewer's comment, the text has been modified to make this idea clearer. 
  2. The meaning of some abbreviations is missing: e.g. line 289 ... mitral flow E/A (?) and line 397 ...CKD... Please double check text e fill the missing quotes. Following the Reviewer's comments, the text has been modified and the abbreviations of E/A and CKD have been included.
  3. Lines 190-196: this refers to a position statement as worldwide grade I obesity is not and indication for Bariatric Surgery and this should be amended. Following the Reviewer's comment, the text has been modified explaining the main conclusions of the position statement regarding BS in patients with class I obesity. 
  4. Lines 428 and following: authors should add and discuss a short paragraph on the so called "obesity paradox", that reports quite the opposite. We fully agree with the Reviewer's comment. Hence, a final paragraph has been included describing the "obesity paradox" at the end of the manuscript.